# Bayesian Active Model Selection
# with an Application to Automated Audiometry

**Jacob R. Gardner**
CS, Cornell University
Ithaca, NY 14850
jrg365@cornell.edu

**Gustavo Malkomes**
CSE, WUSTL
St. Louis, MO 63130
luizgustavo@wustl.edu

**Roman Garnett**
CSE, WUSTL
St. Louis, MO 63130
garnett@wustl.edu

**Kilian Q. Weinberger**
CS, Cornell University
Ithaca, NY 14850
kqw4@cornell.edu

**Dennis Barbour**
BME, WUSTL
St. Louis, MO 63130
dbarbour@wustl.edu

**John P. Cunningham**
Statistics, Columbia University
New York, NY 10027
jpc2181@columbia.edu

## Abstract

We introduce a novel information-theoretic approach for active model selection and demonstrate its effectiveness in a real-world application. Although our method can work with arbitrary models, we focus on actively learning the appropriate structure for Gaussian process (GP) models with arbitrary observation likelihoods. We then apply this framework to rapid screening for noise-induced hearing loss (NIHL), a widespread and preventable disability, if diagnosed early. We construct a GP model for pure-tone audiometric responses of patients with NIHL. Using this and a previously published model for healthy responses, the proposed method is shown to be capable of diagnosing the presence or absence of NIHL with drastically fewer samples than existing approaches. Further, the method is extremely fast and enables the diagnosis to be performed in real time.

## 1   Introduction

Personalized medicine has long been a critical application area for machine learning [1–3], in which automated decision making and diagnosis are key components. Beyond improving quality of life, machine learning in diagnostic settings is particularly important because collecting additional medical data often incurs significant financial burden, time cost, and patient discomfort. In machine learning one often considers this problem to be one of active feature selection: acquiring each new feature (e.g., a blood test) incurs some cost, but will, with hope, better inform diagnosis, treatment, and prognosis. By careful analysis, we may optimize this trade off.

However, many diagnostic settings in medicine do not involve feature selection, but rather involve querying a sample space to discriminate different models describing patient attributes. A particular, clarifying example that motivates this work is *noise-induced hearing loss* (NIHL), a prevalent disorder affecting 26 million working-age adults in the United States alone [4] and affecting over half of workers in particular occupations such as mining and construction. Most tragically, NIHL is entirely preventable with simple, low-cost solutions (e.g., earplugs). The critical requirement for prevention is effective early diagnosis.

To be tested for NIHL, patients must complete a time-consuming audiometric exam that presents a series of tones at various frequencies and intensities; at each tone the patient indicates whether he/she hears the tone [5–7]. From the responses, the clinician infers the patient's audible threshold on a set of discrete frequencies (the *audiogram*); this process requires the delivery of up to hundreds of tones. Audiologists scan the audiogram for a hearing deficit with a characteristic *notch* shape—a

narrow band that can be anywhere in the frequency domain that is indicative of NIHL. Unfortunately, at early stages of the disorder, notches can be small enough that they are undetectable in a standard audiogram, leaving many cases undiagnosed until the condition has become severe. Increasing audiogram resolution would require higher sample counts (more presented tones) and thus only lengthen an already burdensome procedure. We present here a better approach.

Note that the NIHL diagnostic challenge is not one of feature selection (choosing the next test to run and classifying the result), but rather of model selection: is this patient's hearing better described by a normal hearing model, or a notched NIHL model? Here we propose a novel *active model selection* algorithm to make the NIHL diagnosis in as few tones as possible, which directly reflects the time and personnel resources required to make accurate diagnoses in large populations. We note that this is a model-selection problem in the truest sense: a diagnosis corresponds to selecting between two or more sets of indexed probability distributions (models), rather than the more-common misnomer of choosing an index from within a model (i.e., hyperparameter optimization). In the NIHL case this distinction is critical. We are choosing between two models, the set of possible NIHL hearing functions and the set of normal hearing functions. This approach suggests a very different and more direct algorithm than first learning the most likely NIHL function and then accepting or rejecting it as different from normal, the standard approach.

We make the following contributions: first, we design a completely general active-model-selection method based on maximizing the mutual information between the response to a tone and the posterior on the model class. Critically, we develop an analytical approximation of this criterion for Gaussian process (GP) models with arbitrary observation likelihoods, enabling active structure learning for GPs. Second, we extend the work of Gardner et al. [8] (which uses active learning to speed up audiogram inference) to the broader question of identifying *which model*—normal or NIHL—best fits a given patient. Finally, we develop a novel GP prior mean that parameterizes notched hearing loss for NIHL patients. To our knowledge, this is the first publication with an active model-selection approach that does not require updating each model for every candidate point, allowing audiometric diagnosis of NIHL to be performed in real time. Finally, using patient data from a clinical trial, we show empirically that our method typically automatically detects simulated noise-induced hearing loss with fewer than 15 query tones. This is vastly fewer than the number required to infer a conventional audiogram or even an actively learned audiogram [8], highlighting the importance of both the active-learning approach and our focus on model selection.

## 2 Bayesian model selection

We consider supervised learning problems defined on an input space $\mathcal{X}$ and an output space $\mathcal{Y}$. Suppose we are given a set of observed data $\mathcal{D} = (\mathbf{X}, \mathbf{y})$, where $\mathbf{X}$ represents the design matrix of independent variables $\mathbf{x}_i \in \mathcal{X}$ and $\mathbf{y}$ the associated vector of dependent variables $y_i = y(\mathbf{x}_i) \in \mathcal{Y}$.

Let $\mathcal{M}$ be a probabilistic model, and let $\theta$ be an element of the parameter space indexing $\mathcal{M}$. Given a set of observations $\mathcal{D}$, we wish to compute the probability of $\mathcal{M}$ being the correct model to explain $\mathcal{D}$, compared to other models. The key quantity of interest to model selection is the *model evidence*:

$$p(\mathbf{y} \mid \mathbf{X}, \mathcal{M}) = \int p(\mathbf{y} \mid \mathbf{X}, \theta, \mathcal{M})p(\theta \mid \mathcal{M})\,\mathrm{d}\theta, \tag{1}$$

which represents the probability of having generating the observed data under the model, marginalized over $\theta$ to account for all possible members of that model under a prior $p(\theta \mid \mathcal{M})$ [9]. Given a set of $M$ candidate models $\{\mathcal{M}_i\}_{i=1}^{M}$, and the computed evidence for each, we can apply Bayes' rule to compute the posterior probability of each model given the data:

$$p(\mathcal{M} \mid \mathcal{D}) = \frac{p(\mathbf{y} \mid \mathbf{X}, \mathcal{M})p(\mathcal{M})}{p(\mathbf{y} \mid \mathbf{X})} = \frac{p(\mathbf{y} \mid \mathbf{X}, \mathcal{M})p(\mathcal{M})}{\sum_i p(\mathbf{y} \mid \mathbf{X}, \mathcal{M}_i)p(\mathcal{M}_i)}, \tag{2}$$

where $p(\mathcal{M})$ represents the prior probability distribution over the models.

### 2.1 Active Bayesian model selection

Suppose that we have a mechanism for actively selecting new data—choosing $\mathbf{x}^* \in \mathcal{X}$ and observing $y^* = y(\mathbf{x}^*)$—to add to our dataset $\mathcal{D} = (\mathbf{X}, \mathbf{y})$, in order to better distinguish the candidate models

$\{\mathcal{M}_i\}$. After making this observation, we will form an augmented dataset $\mathcal{D}' = \mathcal{D} \cup \{(\mathbf{x}^*, y^*)\}$, from which we can recompute a new model posterior $p(\mathcal{M} \mid \mathcal{D}')$.

An approach motivated by information theory is to select the location maximizing the *mutual information* between the observation value $y^*$ and the unknown model:

$$I(y^*; \mathcal{M} \mid \mathbf{x}^*, \mathcal{D}) = H[\mathcal{M} \mid \mathcal{D}] - \mathbb{E}_{y^*}[H[\mathcal{M} \mid \mathcal{D}']] \tag{3}$$

$$= H[y^* \mid \mathbf{x}^*, \mathcal{D}] - \mathbb{E}_{\mathcal{M}}[H[y^* \mid \mathbf{x}^*, \mathcal{D}, \mathcal{M}]], \tag{4}$$

where $H$ indicates (differential) entropy. Whereas Equation (3) is computationally problematic (involving costly model retraining), the equivalent expression (4) is typically more tractable, has been applied fruitfully in various active-learning settings [10, 11, 8, 12, 13], and requires only computing the differential entropy of the model-marginal predictive distribution:

$$p(y^* \mid \mathbf{x}^*, \mathcal{D}) = \sum_{i=1}^{M} p(y^* \mid \mathbf{x}^*, \mathcal{D}, \mathcal{M}_i) p(\mathcal{M}_i \mid \mathcal{D}) \tag{5}$$

and the model-conditional predictive distributions $\{p(y^* \mid \mathbf{x}^*, \mathcal{D}, \mathcal{M}_i)\}$ with all models trained with the currently available data. In contrast to (3), this does not involve any retraining cost. Although computing the entropy in (5) might be problematic, we note that this is a one-dimensional integral that can easily be resolved with quadrature. Our proposed approach, which we call *Bayesian active model selection* (BAMS) is then to compute, for each candidate location $\mathbf{x}^*$, the mutual information between $y^*$ and the unknown model, and query where this is maximized:

$$\underset{\mathbf{x}^*}{\arg\max}\, I(y^*; \mathcal{M} \mid \mathbf{x}^*, \mathcal{D}). \tag{6}$$

### 2.2 Related work

Although active learning and model selection have been widely investigated, active model selection has received comparatively less attention. Ali et al. [14] proposed an active learning model selection method that requires leave-two-out cross validation when evaluating each candidate $\mathbf{x}^*$, requiring $\mathcal{O}(B^2 M |\mathbf{X}^*|)$ model updates per iteration, where $B$ is the total budget. Kulick et al. [15] also considered an information-theoretic approach to active model selection, suggesting maximizing the expected cross entropy between the current model posterior $p(\mathcal{M} \mid \mathcal{D})$ and the updated distribution $p(\mathcal{M} \mid \mathcal{D}')$. This approach also requires extensive model retraining, with $\mathcal{O}(M |\mathbf{X}^*|)$ model updates per iteration, to estimate this expectation for each candidate. These approaches become prohibitively expensive for real-time applications with large number of candidates. In our audiometric experiments, for example, we consider 10 000 candidate points, expending 1–2 seconds per iteration, whereas these mentioned techniques would take several hours to selected the next point to query.

## 3 Active model selection for Gaussian processes

In the previous section, we proposed a general framework for performing sequential active Bayesian model selection, without making any assumptions about the forms of the models $\{\mathcal{M}_i\}$. Here we will discuss specific details of our proposal when these models represent alternative structures for Gaussian process priors on a latent function.

We assume that our observations are generated via a latent function $f\colon \mathcal{X} \to \mathbb{R}$ with a known observation model $p(\mathbf{y} \mid \mathbf{f})$, where $f_i = f(\mathbf{x}_i)$. A standard nonparametric Bayesian approach with such models is to place a Gaussian process (GP) prior distribution on $f$, $p(f) = \mathcal{GP}(f; \mu, K)$, where $\mu\colon \mathcal{X} \to \mathbb{R}$ is a mean function and $K\colon \mathcal{X}^2 \to R$ is a positive-definite covariance function or kernel [16]. We condition on the observed data to form a posterior distribution $p(f \mid \mathcal{D})$, which is typically an updated Gaussian process (making approximations if necessary). We make predictions at a new input $\mathbf{x}^*$ via the predictive distribution $p(y^* \mid \mathbf{x}^*, \mathcal{D}) = \int p(y^* \mid f^*, \mathcal{D}) p(f^* \mid \mathbf{x}^*, \mathcal{D})\, \mathrm{d}f^*$, where $f^* = f(\mathbf{x}^*)$. The mean and kernel functions are parameterized by hyperparameters that we concatenate into a vector $\theta$, and different choices of these hyperparameters imply that the functions drawn from the GP will have particular frequency, amplitude, and other properties. Together, $\mu$ and $K$ define a model parametrized by the hyperparameters $\theta$. Much attention is paid to learning these hyperparameters in a fixed model class, sometimes under the unfortunate term "model selection."

Note, however, that the *structural* (not hyperparameter) choices made in the mean function $\mu$ and covariance function $K$ themselves are typically done by selecting (often blindly!) from several off-the-shelf solutions (see, for example, [17, 16]; though also see [18, 19]), and this choice has substantial bearing on the resulting functions $f$ we can model. Indeed, in many settings, choosing the nature of plausible functions is precisely the problem of model selection; for example, to decide whether the function has periodic structure, exhibits nonstationarity, etc. Our goal is to automatically and actively decide these structural choices during GP modeling through intelligent sampling.

To connect to our active learning formulation, let $\{\mathcal{M}_i\}$ be a set of Gaussian process models for the latent function $f$. Each model comprises a mean function $\mu_i$, covariance function $K_i$, and associated hyperparameters $\theta_i$. Our approach outlined in Section 2.1 requires the computation of three quantities that are not typically encountered in GP modeling and inference: the hyperparameter posterior $p(\theta \mid \mathcal{D}, \mathcal{M})$, the model evidence $p(\mathbf{y} \mid \mathbf{X}, \mathcal{M})$, and the predictive distribution $p(y^* \mid \mathbf{x}^*, \mathcal{D}, \mathcal{M})$, where we have marginalized over $\theta$ in the latter two quantities. The most-common approaches to GP inference are maximum likelihood–II (MLE) or maximum *a posteriori*–II (MAP) estimation, where we maximize the hyperparameter posterior [20, 16]:[1]

$$\hat{\theta} = \arg\max_\theta \log p(\theta \mid \mathcal{D}, \mathcal{M}) = \arg\max_\theta \log p(\theta \mid \mathcal{M}) + \log(\mathbf{y} \mid \mathbf{X}, \theta, \mathcal{M}). \qquad (7)$$

Typically, predictive distributions and other desired quantities are then reported at the MLE/MAP hyperparameters, implicitly making the assumption that $p(\theta \mid \mathcal{D}, \mathcal{M}) \approx \delta(\hat{\theta})$. Although a computationally convenient choice, this does not account for uncertainty in the hyperparameters, which can be nontrivial with small datasets [9]. Furthermore, accounting correctly for model parameter uncertainty is crucial to model selection, where it naturally introduces a model-complexity penalty. We discuss less-drastic approximations to these quantities below.

## 3.1   Approximating the model evidence and hyperparameter posterior

The model evidence $p(\mathbf{y} \mid \mathbf{X}, \mathcal{M})$ and hyperparameter posterior distribution $p(\theta \mid \mathcal{D}, \mathcal{M})$ are in general intractable for GPs, as there is no conjugate prior distribution $p(\theta \mid \mathcal{M})$ available. Instead, we will use a Laplace approximation, where we make a second-order Taylor expansion of $\log p(\theta \mid \mathcal{D}, \mathcal{M})$ around its mode $\hat{\theta}$ (7). The result is a multivariate Gaussian approximation:

$$p(\theta \mid \mathcal{D}, \mathcal{M}) \approx \mathcal{N}(\theta; \hat{\theta}, \Sigma); \qquad \Sigma^{-1} = -\nabla^2 \log p(\theta \mid \mathcal{D}, \mathcal{M})\big|_{\theta=\hat{\theta}}. \qquad (8)$$

The Laplace approximation also results in an approximation to the model evidence:

$$\log p(\mathbf{y} \mid \mathbf{X}, \mathcal{M}) \approx \log p(\mathbf{y} \mid \mathbf{X}, \hat{\theta}, \mathcal{M}) + \log p(\hat{\theta} \mid \mathcal{M}) - \tfrac{1}{2}\log\det\Sigma^{-1} + \tfrac{d}{2}\log 2\pi, \qquad (9)$$

where $d$ is the dimension of $\theta$ [21, 22]. The Laplace approximation to the model evidence can be interpreted as rewarding explaining the data well while penalizing model complexity. Note that the *Bayesian information criterion* (BIC), commonly used for model selection, can be seen as an approximation to the Laplace approximation [23, 24].

## 3.2   Approximating the predictive distribution

We next consider the predictive distribution:

$$p(y^* \mid \mathbf{x}^*, \mathcal{D}, \mathcal{M}) = \int p(y^* \mid f^*) \underbrace{\int p(f^* \mid \mathbf{x}^*, \mathcal{D}, \theta, \mathcal{M}) p(\theta \mid \mathcal{D}, \mathcal{M}) \, d\theta}_{p(f^* \mid \mathbf{x}^*, \mathcal{D}, \mathcal{M})} \, df^*. \qquad (10)$$

The posterior $p(f^* \mid \mathbf{x}^*, \mathcal{D}, \theta, \mathcal{M})$ in (10) is typically a known Gaussian distribution, derived analytically for Gaussian observation likelihoods or approximately using standard approximate GP inference techniques [25, 26]. However, the integral over $\theta$ in (10) is intractible, even with a Gaussian approximation to the hyperparameter posterior as in (8).

Garnett et al. [11] introduced a mechanism for approximately marginalizing GP hyperparameters (called the MGP), which we will adopt here due to its strong empirical performance. The MGP assumes

that we have a Gaussian approximation to the hyperparameter posterior, $p(\theta \mid \mathcal{D}, \mathcal{M}) \approx \mathcal{N}(\theta; \hat{\theta}, \Sigma)$.[2] We define the posterior predictive mean and variance functions as

$$\mu^*(\theta) = \mathbb{E}[f^* \mid \mathbf{x}^*, \mathcal{D}, \theta, \mathcal{M}]; \qquad \nu^*(\theta) = \mathrm{Var}[f^* \mid \mathbf{x}^*, \mathcal{D}, \theta, \mathcal{M}].$$

The MGP works by making an expansion of the predictive distribution around the posterior mean hyperparameters $\hat{\theta}$. The nature of this expansion is chosen so as to match various derivatives of the true predictive distribution; see [11] for details. The posterior distribution of $f^*$ is approximated by

$$p(f^* \mid \mathbf{x}^*, \mathcal{D}, \mathcal{M}) \approx \mathcal{N}\big(f^*; \mu^*(\hat{\theta}), \sigma^2_{\mathrm{MGP}}\big), \tag{11}$$

where

$$\sigma^2_{\mathrm{MGP}} = \tfrac{4}{3}\nu^*(\hat{\theta}) + \big[\nabla\mu^*(\hat{\theta})\big]^\top \Sigma \big[\nabla\mu^*(\hat{\theta})\big] + \tfrac{1}{3\nu^*(\hat{\theta})}\big[\nabla\nu^*(\hat{\theta})\big]^\top \Sigma \big[\nabla\nu^*(\hat{\theta})\big]. \tag{12}$$

The MGP thus inflates the predictive variance from the the posterior mean hyperparameters $\hat{\theta}$ by a term that is commensurate with the uncertainty in $\theta$, measured by the posterior covariance $\Sigma$, and the dependence of the latent predictive mean and variance on $\theta$, measured by the gradients $\nabla\mu^*$ and $\nabla\nu^*$. With the Gaussian approximation in (11), the integral in (10) now reduces to integrating the observation likelihood against a univariate Gaussian. This integral is often analytic [16] and at worse requires one-dimensional quadrature.

### 3.3   Implementation

Given the development above, we may now efficiently compute an approximation to the BAMS criterion for active GP model selection. Given currently observed data $\mathcal{D}$, for each of our candidate models $\mathcal{M}_i$, we first find the Laplace approximation to the hyperparameter posterior (8) and model evidence (9). Given the approximations to the model evidence, we may compute an approximation to the model posterior (2). Suppose we have a set of candidate points $\mathbf{X}^*$ from which we may select our next point. For each of our models, we compute the MGP approximation (11) to the latent posteriors $\big\{p(\mathbf{f}^* \mid \mathbf{X}^*, \mathcal{D}, \mathcal{M}_i)\big\}$, from which we use standard techniques to compute the predictive distributions $\big\{p(\mathbf{y}^* \mid \mathbf{X}^*, \mathcal{D}, \mathcal{M}_i)\big\}$. Finally, with the ability to compute the differential entropies of these model-conditional predictive distributions, as well as the marginal predictive distribution (5), we may compute the mutual information of each candidate in parallel. See the Appendix for explicit formulas for common likelihoods and a description of general-purpose, reusable code we will release in conjunction with this manuscript to ease implementation.

## 4   Audiometric threshold testing

Standard audiometric tests [5–7] are calibrated such that the average human subject has a 50% chance of hearing a tone at any frequency; this empirical unit of intensity is defined as 0 dB HL. Humans give binary reports (whether or not a tone was heard) in response to stimuli, and these observations are inherently noisy. Typical audiometric tests present tones in a predefined order on a grid, in increments of 5–10 dB HL at each of six octaves. Recently, Gardner et al. [8] demonstrated that Bayesian active learning of a patient's audiometric function significantly improves the state-of-the-art in terms of accuracy and number of stimuli required.

However, learning a patient's entire audiometric function may not always be necessary. Audiometric testing is frequently performed on otherwise young and healthy patients to detect *noise-induced hearing loss* (NIHL). Noise-induced hearing loss occurs when an otherwise healthy individual is habitually subjected to high-intensity sound [27]. This can result in sharp, notch-shaped hearing loss in a narrow (sometimes less than one octave) frequency range. Early detection of NIHL is critical to desirable long-term clinical outcomes, so large-scale screenings of susceptible populations (for example, factory workers), is commonplace [28]. Noise-induced hearing loss is difficult to diagnose with standard audiometry, because a frequency–intensity grid must be very fine to ensure that a notch is detected. The full audiometric test of Gardner et al. [8] may also be inefficient if the only goal of testing is to determine whether a notch is present, as would be the case for large-scale screening.

We cast the detection of noise-induced hearing loss as an active model selection problem. We will describe two Gaussian process models of audiometric functions: a baseline model of normal human

hearing, and a model reflecting NIHL. We then use the BAMS framework introduced above to, as rapidly as possible for a given patient, determine which model best describes his or her hearing.

**Normal-patient model**. To model a healthy patient's audiometric function, we use the model described in [8]. The GP prior proposed in that work combines a constant prior mean $\mu_{\text{healthy}} = c$ (modeling a frequency-independent natural threshold) with a kernel taken to be the sum of two components: a linear covariance in intensity and a squared-exponential covariance in frequency. Let $[i, \phi]$ represent a tone stimulus, with $i$ representing its intensity and $\phi$ its frequency. We define:

$$K\big([i, \phi], [i', \phi']\big) = \alpha i i' + \beta \exp\big(-\tfrac{1}{2\ell^2}|\phi - \phi'|^2\big), \tag{13}$$

where $\alpha, \beta > 0$ weight each component and $\ell > 0$ is a length scale of frequency-dependent random deviations from a constant hearing threshold. This kernel encodes two fundamental properties of human audiologic response. First, hearing is monotonic in intensity. The linear contribution $\alpha i i'$ ensures that the posterior probability of detecting a fixed frequency will be monotonically increasing after conditioning on a few tones. Second, human hearing ability is locally smooth in frequency, because nearby locations in the cochlea are mechanically coupled. The combination of $\mu_{\text{healthy}}$ with $K$ specifies our healthy model $\mathcal{M}_{\text{healthy}}$, with parameters $\theta_{\text{healthy}} = [c, \alpha, \beta, \ell]^\top$.

**Noise-induced hearing loss model**. We extend the model above to create a second GP model reflecting a localized, notch-shaped hearing deficit characteristic of NIHL. We create a novel, flexible prior mean function for this purpose, the parameters of which specify the exact nature of the hearing loss. Our proposed notch mean is:

$$\mu_{\text{NIHL}}(i, \phi) = c - d \mathcal{N}'(\phi; \nu, w^2), \tag{14}$$

where $\mathcal{N}'(\phi; \nu, w)$ denotes the unnormalized normal probability density function with mean $\nu$ and standard deviation $w$, which we scale by a depth parameter $d > 0$ to reflect the prominence of the hearing loss. This contribution results in a localized subtractive notch feature with tunable center, width, and height. We retain a constant offset $c$ to revert to the normal-hearing model outside the vicinity of the localized hearing deficit. Note that we completely model the effect of NIHL on patient responses with this mean notch function; the kernel $K$ above remains appropriate. The combination of $\mu_{\text{NIHL}}$ with $K$ specifies our NIHL model $\mathcal{M}_{\text{NIHL}}$ with, in addition to the parameters of our healthy model, the additional parameters $\theta_{\text{NIHL}} = [\nu, w, d]^\top$.

## 5   Results

To test BAMS on our NIHL detection task, we evaluate our algorithm using audiometric data, comparing to several baselines. From the results of a small-scale clinical trial, we have examples of high-fidelity audiometric functions inferred for several human patients using the method of Gardner et al. [8]. We may use these to simulate audiometric examinations of healthy patients using different methods to select tone presentations. We simulate patients with NIHL by adjusting ground truth inferred from nine healthy patients with in-model samples from our notch mean prior. Recall that high-resolution audiogram data is extremely scarce.

We first took a thorough pure-tone audiometric test of each of nine patients from our trial with normal hearing using 100 samples selected using the algorithm in [8] on the domain $\mathcal{X} = [250, 8000]\,\text{Hz} \times [-10, 80]\,\text{dB HL},$[3] typical ranges for audiometric testing [6]. We inferred the audiometric function over the entire domain from the measured responses, using the healthy-patient GP model $\mathcal{M}_{\text{healthy}}$ with parameters learned via MLE–II inference. The observation model was $p(y = 1 \mid f) = \Phi(f)$, where $\Phi$ is the standard normal CDF, and approximate GP inference was performed via a Laplace approximation. We then used the approximate GP posterior $p(f \mid \mathcal{D}, \hat{\theta}, \mathcal{M}_{\text{healthy}})$ for this patient as ground-truth for simulating a healthy patient's responses. The posterior probability of tone detection learned from one patient is shown in the background of Figure 1(a). We simulated a healthy patient's response to a given query tone $\mathbf{x}^* = [i^*, \phi^*]$ by sampling a conditionally independent Bernoulli random variable with parameter $p(y^* = 1 \mid \mathbf{x}^*, \mathcal{D}, \hat{\theta}, \mathcal{M}_{\text{healthy}})$.

We simulated a patient with NIHL by then drawing notch parameters (the parameters of (14)) from an expert-informed prior, adding the corresponding notch to the learned healthy ground-truth latent mean, recomputing the detection probabilities, and proceeding as above. Example NIHL ground-truth detection probabilities generated in this manner are depicted in the background of Figure 1(b).

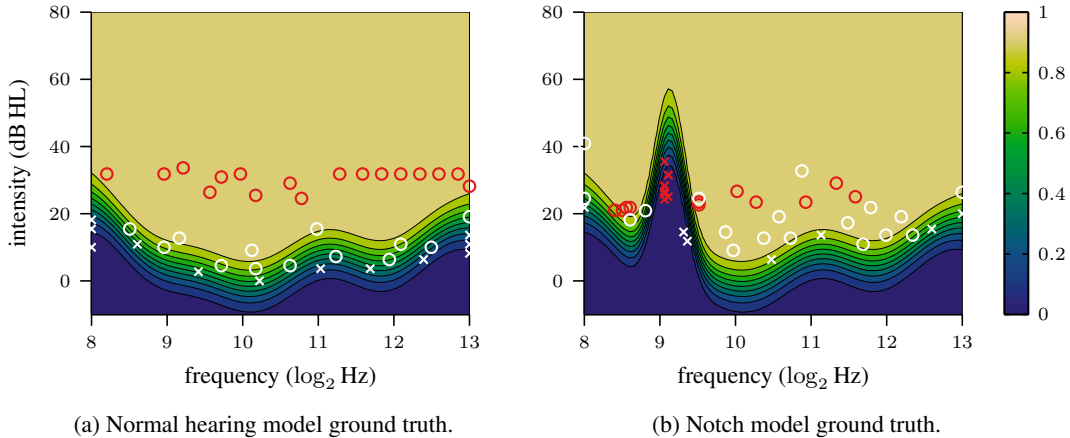

(a) Normal hearing model ground truth.　　　　(b) Notch model ground truth.

Figure 1: Samples selected by BAMS (red) and the method of Gardner et al. [8] (white) when run on (a) the normal-hearing ground truth, and (b), the NIHL model ground truth. Contours denote probability of detection at 10% intervals. Circles indicate presentations that were heard by the simulated patient; exes indicate presentations that were not heard by the simulated patient.

## 5.1 Diagnosing NIHL

To test our active model-selection approach to diagnosing NIHL, we simulated a series of audiometric tests, selecting tones using three alternatives: BAMS, the algorithm of [8], and random sampling.[4] Each algorithm shared a candidate set of 10 000 quasirandom tones $\mathbf{X}^*$ generated using a scrambled Halton set so as to densely cover the two-dimensional search space. We simulated nine healthy patients and a total of 27 patients exhibiting a range of NIHL presentations, using independent draws from our notch mean prior in the latter case. For each audiometric test simulation, we initialized with five random tones, then allowed each algorithm to actively select a maximum of 25 additional tones, a very small fraction of the hundreds typically used in a regular audiometric test. We repeated this procedure for each of our nine healthy patients using the normal-patient ground-truth model. We further simulated, for each patient, three separate presentations of NIHL as described above. We plot the posterior probability of the correct model after each iteration for each method in Figure 2.

In all runs with both ground-truth models, BAMS was able to rapidly achieve greater than 99% confidence in the correct model without expending the entire budget. Although all methods correctly inferred high healthy posterior probability for the healthy patient, BAMS wass more confident. For the NIHL patients, neither baseline inferred the correct model, whereas BAMS rarely required more than 15 actively chosen samples to confidently make the correct diagnosis. Note that, when BAMS was used on NIHL patients, there was often an initial period during which the healthy model was favored, followed by a rapid shift towards the correct model. This is because our method penalizes the increased complexity of the notch model until sufficient evidence for a notch is acquired.

Figure 1 shows the samples selected by BAMS for typical healthy and NIHL patients. The fundamental strategy employed by BAMS in this application is logical: it samples in a row of relatively high-intensity tones. The intuition for this design is that failure to recognize a normally heard, high-intensity sound is strong evidence of a notch deficit. Once the notch has been found (Figure 1(b)), BAMS continues to sample within the notch to confirm its existence and rule out the possibility of the miss (tone not heard) being due to the stochasticity of the process. Once satisfied, the BAMS approach then samples on the periphery of the notch to further solidify its belief.

The BAMS algorithm sequentially makes observations where the healthy and NIHL model disagree the most, typically in the top-center of the MAP notch location. The exact intensity at which BAMS samples is determined by the prior over the notch-depth parameter $d$. When we changed the notch depth prior to support shallower or deeper notches (data not shown), BAMS sampled at lower or

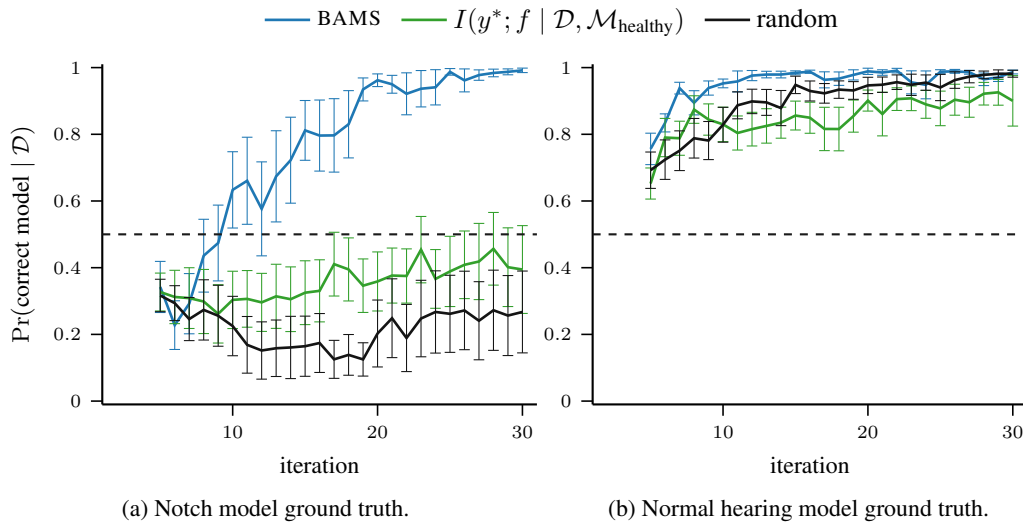

| BAMS | $I(y^*; f \mid \mathcal{D}, \mathcal{M}_{\text{healthy}})$ | random |

(a) Notch model ground truth.

(b) Normal hearing model ground truth.

Figure 2: Posterior probability of the correct model as a function of iteration number.

higher intensities, respectively, to continue to maximize model disagreement. Similarly, the spacing between samples is controlled by the prior over the notch-width parameter $w$.

Finally, it is worth emphasizing the stark difference between the sampling pattern of BAMS and the audiometric tests of [8]; see Figure 1. Indeed, when the goal is learning the patient's audiometric function, the audiometric testing algorithm proposed in that work typically has a very good estimate after 20 samples. However, when using BAMS, the primary goal is to detect or rule out NIHL. As a result, the samples selected by BAMS reveal little about the nuances of the patient's audiometric function, while being highly informative about the correct model to explain the data. This is precisely the tradeoff one seeks in a large-scale diagnostic setting, highlighting the critical importance of focusing on the model-selection problem directly.

## 6  Conclusion

We introduced a novel information-theoretic approach for active model selection, *Bayesian active model selection*, and successfully applied it to rapid screening for noise-induced hearing loss. Our method for active model selection does not require model retraining to evaluate candidate points, making it more feasible than previous approaches. Further, we provided an effective and efficient analytic approximation to our criterion that can be used for automatically learning the model class of Gaussian processes with arbitrary observation likelihoods, a rich and commonly used class of potential models.

### Acknowledgments

This material is based upon work supported by the National Science Foundation (NSF) under award number IIA-1355406. Additionally, JRG and KQW are supported by NSF grants IIS-1525919, IIS-1550179, and EFMA-1137211; GM is supported by CAPES/BR; DB acknowledges NIH grant R01-DC009215 as well as the CIMIT; JPC acknowledges the Sloan Foundation.

## Footnotes

[1]Using a noninformative prior $p(\theta \mid \mathcal{M}) \propto 1$ in the case of maximum likelihood.

[2]This is arbitrary and need not be the Laplace approximation in (8), so this is a slight abuse of notation.

[3]Inference was done in log-frequency domain.

[4]We also compared with uncertainty sampling and query by committee (QBC); the performance was comparable to random sampling and is omitted for clarity.

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
