[Supplementary Material]

# Bayesian Active Model Selection
# Supplementary Material

**Jacob R. Gardner**
CS, Cornell University
Ithaca, NY 14850
jrg365@cornell.edu

**Gustavo Malkomes**
CSE, WUSTL
St. Louis, MO 63130
luizgustavo@wustl.edu

**Roman Garnett**
CSE, WUSTL
St. Louis, MO 63130
garnett@wustl.edu

**Kilian Q. Weinberger**
CS, Cornell University
Ithaca, NY 14850
kqw4@cornell.edu

**Dennis Barbour**
BME, WUSTL
St. Louis, MO 63130
dbarbour@wustl.edu

**John P. Cunningham**
Statistics, Columbia University
New York, NY 10027
jpc2181@columbia.edu

## 1   Computational details for common observation likelihoods

Here we give further details for computing (4) with common observation likelihoods.

### 1.1   Regression with Gaussian noise

For regression problems with zero-mean homoskedastic Gaussian noise with variance $\sigma_n^2$, we have

$$p(y \mid f) = \mathcal{N}(y; f, \sigma_n^2).$$

We may integrate this against a Gaussian distribution on the latent value $f$ (such as that resulting from the MGP approximation (11)) to find the predictive distribution. Suppose $p(f \mid \mathbf{x}, \mathcal{D}) = \mathcal{N}(f; \mu, \sigma^2)$. Then

$$p(y \mid \mathbf{x}, \mathcal{D}) = \mathcal{N}(y; \mu, \sigma^2 + \sigma_n^2).$$

The model-conditional predictive distributions $\{p(y^* \mid \mathbf{X}^*, \mathcal{D}, \mathcal{M}_i)\}$ are thus Gaussian under the MGP approximation:

$$p(y^* \mid \mathbf{x}^*, \mathcal{D}, \mathcal{M}_i) = \mathcal{N}\big(y^*; \mu_i^*, (\nu_{\text{MGP}}^*)_i + \sigma_n^2\big).$$

The differential entropy of a one-dimensional Gaussian is

$$H\big[\mathcal{N}(y; \mu, \sigma^2)\big] = \tfrac{1}{2}\log(2\pi e \sigma^2);$$

with these results and an approximation to the model posterior, we may compute the second term in (4). The marginal predictive distribution $p(y^* \mid \mathbf{x}^*, \mathcal{D})$ is now a mixture of Gaussians weighted by the approximate model posterior. The differential entropy of a mixture of Gaussians does not have a closed form, so for the first term in (4), we must resort to (one-dimensional) quadrature.

Note that we may also treat the noise variance $\sigma_n^2$ as a model-dependent hyperparameter and use the MGP to approximately marginalize it as well, which changes the above only slightly. An extension to heteroskedastic noise is also trivial.

### 1.2   Probit regression

For binary classification problems with a probit likelihood, we have

$$\Pr(y = 1 \mid f) = \Phi(f),$$

where $\Phi$ is the univariate standard normal CDF. We may integrate this against a Gaussian distribution on the latent value $f$ to find the predictive distribution. Suppose $p(f \mid \mathbf{x}, \mathcal{D}) = \mathcal{N}(f; \mu, \sigma^2)$. Then

$$\Pr(y = 1 \mid \mathbf{x}, \mathcal{D}) = \int \Phi(f)\mathcal{N}(f; \mu, \sigma^2)\, \mathrm{d}f = \Phi\left(\frac{\mu}{\sqrt{1 + \sigma^2}}\right).$$

The model-conditional predictive distributions $\left\{ p(y^* \mid \mathbf{X}^*, \mathcal{D}, \mathcal{M}_i) \right\}$ are thus Bernoulli distributions under the MGP approximation:

$$p(y^* \mid \mathbf{x}^*, \mathcal{D}, \mathcal{M}_i) = \mathcal{B}\left( \Phi\left( \frac{\mu_i^*}{\sqrt{1 + (\nu_{\mathrm{MGP}}^*)_i}} \right) \right).$$

The differential entropy of a Bernoulli distribution with success probability $p$ is given by the Bernoulli entropy function $h$:

$$H\big[\mathcal{B}(p)\big] = h(p) = -p \log p - (1 - p) \log(1 - p);$$

with these results and an approximation to the model posterior, we may compute the second term in (4). The marginal predictive distribution $p(y^* \mid \mathbf{x}^*, \mathcal{D})$ is now a mixture of Bernoullis weighted by the approximate model posterior. Bernoulli distributions are closed under taking mixtures, and therefore in this case we may compute both terms in (4) without resorting to quadrature.

## 2   Description of general-purpose code

Finding a Laplace approximation to a GP hyperparameter posterior $p(\theta \mid \mathcal{D}, \mathcal{M})$ (8) requires evaluating the Hessian of the log posterior at the mode, a somewhat atypical computation. MATLAB code for calculating this Hessian for exact inference (with a Gaussian observation likelihood) has been previously published;[1] along with this manuscript, we will provide general-purpose MATLAB code (built on top of the popular GPML toolkit [1]) for calculating this Hessian for arbitrary observation models using the Laplace approximation for approximate GP inference [2].

Computing the MGP predictive variance (12) requires the gradient of the latent predictive mean and variance with respect to the hyperparameters, another uncommon computation. We will also provide code computing these gradients both for exact inference and approximate inference using the Laplace approximation, allowing the MGP to be used for arbitrary observation likelihoods. With this and the above-mentioned code, our entire scheme for active Bayesian model selection with GPs can be implemented with minimal effort on top of the popular GPML toolbox.

## Footnotes

[1] https://github.com/rmgarnett/gpml_extensions