[Reviews · NeurIPS 2015]

Submitted by Assigned_Reviewer_1

# Summary

The authors introduce a new method for actively selecting the model that best fits a dataset. Contrary to active learning, where the next learning point is chosen to get a better estimate of the model hyperparameters, this methods selects the next point to better distinguish between a set of models. Similar active model selection techniques exist, but they need to retrain each model for each new data point to evaluate. The strength of the author's method is that is only requires to evaluate the predictive distributions of models, without retraining.

They propose to apply this method to detect noise-induced hearing loss. The traditional way of screening for NIHL involves testing a wide range of intensities and frequencies, which is time consuming. The authors show that with their method, the number of tests to be run could be drastically decreased, reducing the cost of large-scale screenings for NIHL.

# Quality

The paper is technically sound. Although not detailed, the derivations make sense. However, there is no theoretical justification or discussion, and the empirical evaluation is weak. The authors only compare their active model selection scheme with a "traditional" active learning algorithm, and a random baseline, on simulated data generated from their model (possibly biasing the results). Ideally, it should also be evaluated on real data with true examples of NIHL (not generated from the model), and also compared with other active model selection methods. Moreover, some key issues, such as defining the set of candidate points or candidate models, are not discussed at all.

# Clarity

The paper is very well written. The structure is clear, and the relevance and usefulness of the proposed method are well introduced. The application is particularly well explained. A pleasure to read!

# Originality

As stated by the authors, the problem of active model selection is less well-studied than that of active learning. The proposed method is a variation of exiting ones (maximizing mutual information instead of expected cross entropy, for example), that relies on various existing approximations for its implementation. Its application is also original, and not well studied.

# Significance

The method gives a significant complexity improvement over existing ones, as it does not require retraining the models for each new candidate. It enables real-time application of the method, which is key to some problems. It also allows to reduce significantly the number of testing points required for NIHL screening, which would have a significant impact on large-scale screening, reducing costs.

# Pros

* Well written and clear * New method for active model selection, that does not require retraining models * Significant gains (fewer samples required) for an important and practical application

# Cons

* No theoretical analysis/discussion of the new method * Weak empirical evaluation * No comparison with other active model selection techniques * Evaluation on data simulated from their own model, thus possibly biasing results * No discussion of the problem of generating the set of candidate points or models
Summary: The novel active model selection approach the authors propose seems promising, and the application useful. However, the evaluation is not convincing, and their is no discussion of some key issues like selecting the candidate locations and models.

Submitted by Assigned_Reviewer_2

This paper presents a method for active model selection using a mutual information criterion. The main technical contribution is a way to approximate the MI criterion without having to re-fit the model for each candidate. The method is evaluated on the task of detecting noise-induced hearing loss with as few sound queries as possible.

Overall, this paper presents a promising and well thought out approach to the active model selection problem. The paper is readable and motivates the design choices well. The NIHL detection task is an interesting use of active model selection, and its introduction is a useful contribution of the paper.

One thing missing from the paper is experimental comparisons against prior approaches to active model selection, either on the NIHL task or on benchmarks where other methods have been successfully applied. In the experiment, the baseline [8] shows a pathology (focusing on fitting the more likely model) which other active model selection methods would presumably correct, so it would be interesting to see how the different approaches compare.

The NIHL task is a bit limited in that the positive examples are simulated from the model, which could lead model based methods to perform unrealistically well. This may be inevitable in the medical domain, though, and I don't immediately see a better way to set up the experiment.

Summary: Overall, I would recommend acceptance because the paper is well written, the methods seem novel and well motivated, and the paper introduces a real-world medical task which may benefit from the proposed approach. The main thing missing is comparisons to other active model selection approaches. I'm not an expert on active learning, so I can't speak with confidence about the relationship to prior work.

Submitted by Assigned_Reviewer_3

I had a couple of main comments/questions.

I would have liked to seen empirical evaluation of the approach on data from NIHL subjects (rather than on modified normal hearing data) for two main reasons. First, it would have been especially useful to see empirical justification for the NIHL mean function and the priors on the various hyper parameters which presumably are key to getting the method to perform sensibly e.g. so a NIHL notch is not explained by the SE fluctuations in a normal audiogram. Second, I have slight concerns about the robustness of the method and exposing it to real data would be a sensible test.

Is model comparison is really the most sensible approach to diagnosis -- doesn't everyone have some degree of noise induced hearing loss and so shouldn't diagnosis correspond to identifying where on a spectrum the individual is? Would it then not be more sensible/sufficient to do inference in the NIHL model, define the mutual information gain in terms of the parameter 'd', and base diagnosis on the magnitude of this inferred parameter, rather than using a discrete mixture model?
Summary: I like the paper and vote of acceptance; the application of active learning approaches to automated audiometry is very sensible. The paper is well written, clear and the model choices and approximations well justified.

Submitted by Assigned_Reviewer_4

This work introduces a new criterion for sequential experimental design with the goal of model selection. The author(s) applies this algorithm to an audiometric experiment consisting of detecting noise-induced hearing loss. The experiment produces good results.

Quality and clarity: This paper is very clear and very well written. It nicely outlines the steps necessary in approximating model evidence, hyperparameter posterior and predictive distributions. Minor points: on line 134, X* is mentioned before it is defined (actually, I don't believe it is ever defined) and on line 342, "exes" -> "crosses."

Originality: The author(s)'s work applies a trick similar to that of [12], using the symmetry of mutual information to turn an intractable quantity into a one-dimensional intractable quantity that can be approximated relatively easy. They apply this technique to the problem of active model selection, which is novel.

Significance: The problem of model selection is obviously an important one. The particular application discussed in the paper of selecting among different GP priors is a prime example. Unfortunately, the example provided was simple and included only selecting among prior means and not the covariance kernel, which is often the more difficult to choose. In addition, the first model (healthy) is included in the NIHL model (as d -> 0). Therefore it begs the question of whether one could simply use the more complex model and infer the hyperparameters, and then examine the magnitude of d and/or w. Though I do not doubt the potential applications of this method, I do question how convincing this particular one is.

Minor points: - Section 3: it is worth at least mentioning the hyperparameter marginalization via MCMC technique around equation (7).
Summary: This is a very nice paper about a Bayesian method for active model selection. The techniques used in the method are explained at a very appropriate level of detail with useful references for further reading. Though compelling, in my opinion, the application presented failed to illustrate the strength of the method due to concerns I raise under the "significance" heading below.

Author Feedback
Author rebuttal: First, we want to thank all reviewers for their time and reviews. Answers to specific comments and questions are below.

Evaluation on real patients: We agree that evaluating our method on real patients is an important next step, and we are working with physicians who are planning to use our approach in clinical studies as soon as possible. We are hoping to publish the empirical results in a medical journal in the near future.

Comparison with other active model selection techniques: the previous active model selection approaches are impractical due to extensive model-retraining; see the running time analysis discussed in Section 2.2. For instance--all other costs aside--if just updating all of the model parameters took only 1 second, selecting a single tone from a set of 20,000 candidates points (as we did) would take at least 5 hours. Our approach (as well as the chosen baselines) takes less than 2 seconds overall.

Using model selection: In principle, detecting NIHL by doing inference on the "d" hyperparameter is an interesting idea. Currently, our expert informed prior over "d" essentially excludes the possibility of high posterior weight on d=0. A multimodal prior (e.g., a mixture of our expert "notch" prior with a Gaussian near d=0) would therefore be required. At a high level, this strongly resembles what we already do, as the mixture weights can essentially be viewed as the model posterior. However, inference is less tractable, as a Gaussian approximation to the hyperparameter posterior is less appropriate. Our framework overcomes this problem and is more general, as not all models can be easily rewritten as parameterizations of other models.

Choosing candidate models/points: 20000 points were chosen from a scrambled Halton set, so as to densely cover the two-dimensional search space. In the NIHL setting, we only need two models: a "normal" hearing model and a NIHL model. The normal hearing model is taken from existing machine learning literature [8], and has further been evaluated in clinical trials [A]. The NIHL model was chosen in collaboration with physicians. For more general model selection settings we would recommend adapting the strategies in [17,18].

MCMC: Note that [11] did an MCMC study on the approximation used in section 3.2. In our setting, we have strict time requirements for choosing each candidate point that make the use of MCMC undesirable.

[A] X. D. Song, B. M. Wallace, J. R. Gardner, N. M. Ledbetter, K. Q. Weinberger, and D. L Barbour. Fast, continuous audiogram estimation using machine learning. Ear and Hearing, 2015.